# Enhancement and Numerical Assessment of Novel SARS-CoV-2 Virus Transmission Model

## Abstract

Recent pandemic of the coronavirus started in December 2019, which has affected almost all groups of humankind. In this regard, accurate epidemic models are not only crucial for demonstrating the mitigation of the current pandemic but also helpful for forecasting their future dynamics. In this work, we propose a model for SARS-CoV-2 virus transmission to forecast the temporal dynamics of the novel coronavirus disease by considering the characteristics of the disease and the recent literature. Due to the nondeterministic and stochastic nature of the novel-coronavirus disease, we present the model with the aid of stochastic differential equations by considering two infectious phases: pre-symptomatic and symptomatic, because both are significant in the spread of SARS-CoV-2 virus transmission. We ensure that the model is well-posed and identify the necessary conditions for disease eradication by proving the existence, uniqueness, and extinction analysis. The efficacy of the model and the importance of the current study are demonstrated using the actual data. Finally, the model will be simulated using Euler-Maruyama and Milstein's numerical schemes to support the theoretical findings and show the significance of the results obtained.

## 1 Introduction

Diseases are mainly categorized into two groups: infectious and non-infectious diseases. Infectious are those caused by viruses, fungi, parasites, and bacteria, and usually transferred in numerous ways while causing fifty thousand deaths approximately every day all over the world. Some infectious diseases can be directly communicated, while many transfer indirectly. Infectious diseases like hepatitis B, seasonal influenza, HIV (human immunodeficiency virus), Middle East Respiratory Syndrome (MERS), and SARS-CoV-2 are major health issues, affecting millions of populations around the globe (Altamimi et al., 2020; Holmdahl & Buckee, 2020; Mann & Roberts, 2011; Reich et al., 2019; Park et al., 2021). SARS-CoV is a family of viruses that usually cause illnesses such as MERS and severe respiratory syndrome coronavirus (Syed, 2020). Coronaviruses are zoonotic diseases, and the novel one is a new strain known as the SARS-CoV-2 virus, which broke out in 2019 and spread throughout the world (Rabaan et al., 2020). Bats are the most plausible ecological reservoirs for SARS-CoV-2, but it's also possible that the virus infected humans via an intermediate animal host. This intermediate animal host could be an unidentified domesticated food animal, a wild animal, or a domesticated wild animal. Millions of people are infected and face consequences due to the novel disease of coronavirus (Shereen et al., 2020). Novel coronavirus transmits from one person to another by direct contact with an infected individual and indirect with objects used by the infected person (Chan et al., 2020). A novel coronavirus disease has multiple phases of infections, pre-symptomatic, asymptomatic, environmental, and symptomatic (He et al., 2020). Especially, the pre-symptomatic and symptomatic phases are very significant because 47% and 38% of cases are reported respectively, by contact with these individuals (Ferretti et al., 2020). Generally, pre-asymptomatic individuals have no symptoms while transmitting the disease to others. Therefore, the immigration of pre-symptomatic and symptomatic patients from one place to another place leads to a major source of novel coronavirus transmission. So, most countries around the globe restricted air traffic and announced a lockdown to use the precautionary measure to minimize human lives as much as possible. Also, every country tried to reduce unnecessary traveling, which helped in the reduction of the newly reported cases. Many countries are badly affected by the pandemic of SARS-CoV-2. Moreover, the economy of different countries has been influenced by the novel deadly virus

badly. Many organizations and companies have stopped their production. As a result, the ratio of unemployment as well as poverty has increased in various countries. Besides, the health systems of many developed and powerful countries collapsed due to the consequences of the SARS-CoV-2 virus.

Mathematical modeling is one of the best tools to show disease mitigation and design control mechanism. The epidemiology of infectious disease has a rich literature, see for instance, Mann & Roberts (2011); Hattaf et al. (2012); Shi et al. (2015); Alaa & van der Schaar (2019); Kamarthi et al. (2021); Yin et al. (2021). Various models have been used extensively by researchers to study the temporal dynamics of different infectious diseases (Wang et al., 2014; Khan et al., 2018). Moreover, many mathematicians and biologists also studied the dynamical behaviors of SARS-CoV-2 transmission. More precisely, a model study has been reported to forecast the spread of a novel coronavirus outbreak in Wuhan (Wu et al., 2020). Guo et al. (2020) studied the prediction of host and infectivity of novel diseases using deep learning algorithm. Selvam et al. (2021) analyzed the spread of corona virus diseases using mathematical modeling and performed stability analysis. Further, a spatial-temporal model has been formulated to study the dynamic risk assessment of SARS-CoV-2 accounting for community virus exposure (Chen et al., 2021). Similarly, many other studies have been reported by various authors to forecast the pandemic trend and control of novel disease (SARS-CoV-2) transmissions (see for more details, Arik et al. (2020); Kucharski et al. (2020); Wang et al. (2020); Flaxman et al. (2020); Chen et al. (2021); Zhang et al. (2021)). Very recently, a stochastic epidemiological model has been analyzed for the dynamics of novel disease of corona virus by Khan et al. (2021). Models with appropriate structure and accurate dynamics are not only important to show disease mitigation but also to forecast the future dynamics of the disease, and thus are helpful for public health planning. Indeed, the above-reported studies yielded some interesting outputs. However, their main objectives are to forecast disease dynamics using deterministic differential equations. While the model related to the stochastic analysis of SARS-CoV-2 virus transmission is reported in Khan et al. (2021), many factors are to be improved. For example, they considered the random fluctuation only in the disease transmission rate, but the other parameters such as mobility, vaccination, the occurrence of death, etc. also have stochastic nature. Moreover, the classification of various infection phases of SARS-CoV-2 virus disease has not been considered, such that important roles played by pre-symptomatic and symptomatic individuals were neglected in the pandemic trend of novel corona virus disease. In fact, a small number of pre-symptomatic individuals will lead to a major outbreak, because they have no symptoms while transmitting the disease to others. Further, the pandemic of SARS-CoV-2 rises due to human interaction, but initial sources of transmission were a reservoir that has been ignored.

Our goal is to enhance the model reported in Khan et al. (2021) by incorporating the missing parameters and characteristics of SARS-CoV-2 virus disease that can influence the disease transmission. To this end, we use various sources of randomness using different Brownian motions in each population group to include the stochastic effect in every parameter as well as in every group of population. We divide the total infected population into two sub-classes, namely pre-symptomatic and symptomatic, according to the characteristics of the disease. We will also assume that both the pre-symptomatic and symptomatic individuals will contribute to producing the reservoirs. To do this, first, we formulate the model and discuss its biological and mathematical feasibility to show the well-posedness of the problem. For this purpose, we will use the combination of stochastic Lyapunov function theory and the Itô formula. Further, the disease extinction of the model will be discussed to find the conditions for disease elimination. We then develop the algorithms for the proposed epidemic problem to discuss the numerical assessment and verify our analytical findings by using Euler-Maruyama and Milstein's methods. To show the effectiveness and justify the proposed epidemic problem, we fit the model to real data of the SARS-CoV-2 virus reported in the United Arab Emirates in the period of March, 21st 2022 to Jun, 21st 2022. Finally, we compare our results to show the significance of the model solution.

## 2 PRELIMINARIES

In this section, we introduce some of the fundamental concepts and notations that will be helpful in getting our main results, including a multidimensional Itô formula, the strong law of large numbers, and some other results.

Let $\mathcal{R}$ be the set of real numbers, $\mathcal{R}^n$ the space of n-tuple, and $\mathcal{R}_+^n$ the set of n-tuple with non-negative entries. $(\Theta, \mathcal{F}_T, (\mathcal{F}_t)_{t \in [0,T]}, P)$ represents a filtered probability space $(\Theta, \mathcal{F}_T, P)$ with a right-continuous and complete filtration $\mathbb{F} = (\mathcal{F}_t)_{t \in [0,T]}$, i.e., $\mathbb{F}^+ = \mathbb{F}$ and each $\mathcal{F}_t \in \mathbb{F}$ contains all $P$-null spaces (Klenke, 2007). If $f(x)$ is a function over $[a, b]$, then the mean value of $f(x)$ is defined as

$$\langle f(x) \rangle := \frac{1}{b-a} \int_a^b f(x) dx.$$

In addition, $\mathcal{C}^2(\mathcal{R}^n)$ denotes the set of all real-valued and twice continuously differentiable functions.

**Lemma 1** *(Kuo, 2006) Let $a = (\alpha_1, \ldots, \alpha_n)$ and $b = (\beta_1, \ldots, \beta_n)$ represent the $n$-dimensional square-integrable adapted processes. We assume that $\mathcal{X} = (\mathcal{X}_1, \ldots, \mathcal{X}_n)$, where $\mathcal{X}_k$ for $k \in \{1, \ldots, n\}$ is driven by the stochastic differential equation (SDE) with standard Brownian motion $\mathcal{B}(t)$, i.e.,*

$$d\mathcal{X}_k(t) = a_k dt + \beta_k d\mathcal{B}(t), \quad \mathcal{X}_k(0) \in \mathcal{R}.$$

*Let $\mathcal{H} \in \mathcal{C}^2(\mathcal{R}^n)$, then*

$$d\mathcal{H}(\mathcal{X}(t)) = \sum_{k=1}^n \frac{\partial \mathcal{H}(\mathcal{X}(t))}{\partial x_k} d\mathcal{X}_k(t) + \sum_{k,l=1}^n \frac{1}{2} \frac{\partial^2 \mathcal{H}(\mathcal{X}(t))}{\partial x^k x^l} d\langle \mathcal{X}_k(t), \mathcal{X}_l(t) \rangle,$$

*where $\langle, \rangle$ represents the inner product between two functions in $\mathcal{C}^2(\mathcal{R}^n)$.*

**Lemma 2** *(Birkel, 1988) Let $\sigma$ be the intensity of environmental fluctuation and $B(t)$ the standard Brownian motion, and assume that $N = \{N_t\}_{t \geq 0}$ is a continuous real-valued martingale defined by $N(t) = \int_0^t \sigma d\mathcal{B}(s)$ and $\langle N, N \rangle_t = \int_0^t \sigma^2 dt$, vanishes at $t = 0$, then*

$$\lim_{t \to \infty} \langle N, N \rangle_t = \infty, \quad \textit{implies} \quad \lim_{t \to \infty} \frac{N}{\langle N, N \rangle_t} = 0.$$

*Also*

$$\limsup_{t \to \infty} \frac{\langle N, N \rangle_t}{t} < 0, \quad \textit{implies} \quad \lim_{t \to \infty} \frac{N_t}{t} = 0.$$

## 3 MODEL FORMULATION AND ANALYSIS

In this section, we will discuss the enhancement of the model reported in Khan et al. (2021). For this, let us assume that $(\Theta, \mathcal{F}_T, (\mathcal{F}_t)_{t \in [0,T]}, P)$, on this space lives a 5-D Brownian motion $\mathcal{W} := \{\mathcal{W}(t) : t \in [0,T]\}$ with $\mathcal{W}(t) := \{\mathcal{W}_i(t) : i = 1, ..., 5\}$. Moreover, the natural filtration $(\mathcal{F}_t)_{t \in [0,T]}$ is assumed to be generated by the Brownian motion $\mathcal{W}$. Further, it could be noted that the spread of SARS-CoV-2 disease is uncertain due to its novel nature. Thus, it transmits from human to human as well as from reservoir to human while not uniform everywhere. We, therefore, assume that the various compartments of the population have stochastic nature driven by different randomness sources $\mathcal{B}_i(t)$, $i = 1, \ldots, 5$. The proposed epidemiological model assumes that the variation in all groups of population is connected to various sources of information represented by $(\mathcal{F}_t)_{t \geq 0}$, where $\mathcal{F}_t := \sigma(\mathcal{B}(t))$ symbolizes the $\sigma-$algebra. We then assume the random fluctuation in every group of the population and formulate the model based on available information about the biological process through which the SARS-CoV-2 virus spreads. The multiple infection phases and reservoirs play an important role in the novel disease transmission. The pre-symptomatic and symptomatic populations are especially very significant. We divide the entire host population into various classes: $s =$ susceptible, $a =$ pre-symptomatic, $b =$ symptomatic, $r =$ recovered, and $m =$ reservoir. The notion $s$ defines the un-infected population at time $t$, but has a chance to get infected at time $t + \Delta t$, where $\Delta t$ is the small increment in time. Similarly, the pre-symptomatic are those who are infected while having no symptoms and are very significant in SARS-CoV-2 transmission because of its novel nature, while the infected individuals showing symptoms are known as the symptomatic population. The individuals who become healthy after the infection are the recovered ones. We assume that all newborn babies, as well as immigrants from other populations, are susceptible. Since still there is not enough evidence on the vertical transmission of the virus from the pregnant

woman to the newborn baby, however, immigrants may or may not be infected. In the present case, for the sake of mathematical simplicity, we assume that the immigrants are virus-free. Since the population is assumed to be homogeneously mixed, the successful transmission of SARS-CoV-2 leads the susceptible individuals $s$ into the pre-symptomatic or symptomatic with probability $p$ and $(1-p)$, respectively. Both the infected population (pre-symptomatic and symptomatic) and the reservoir are responsible for disease transmission. We specify two modes of recovery based on the characteristic of the disease. First, a fraction, denoted by $\rho$, of the pre-symptomatic population may recover directly with $q$ probability and enter the recovered class $r$ in case the individual has a strong immune system, while those who develop symptoms will lead to the symptomatic compartment with $(1-q)$ probability. Second, the individual leaves the symptomatic class only after they have fully recovered by taking proper treatment and move to the recovered class $r$ or die. Based on the effectiveness of the vaccine, the $v$-th portion of the susceptible population will move to the recovered class directly. The ratio of the reservoirs contributing from infected (both the pre-symptomatic and symptomatic) population are respectively denoted by $\alpha_1$ and $\alpha_2$, while the removal rate for the reservoir is $\eta$. Further, the geometry of the proposed problem is depicted in Figure 1 and so the

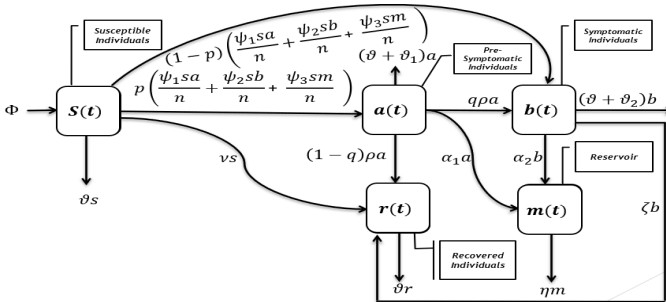

Figure 1: The schematic process of the SARS-CoV-2 virus transmission model (1)

enhanced stochastic epidemiological model looks like this:

$$ds(t) = \left\{ \Phi - \frac{\psi_1 s(t)a(t)}{n(t)} - \frac{\psi_2 b(t)s(t)}{n(t)} - \frac{\psi_3 s(t)m(t)}{n(t)} - (v+\vartheta)s(t) \right\} dt + \eta_1 s(t)d\mathcal{B}_1(t),$$

$$da(t) = \left\{ \left( \frac{\psi_1 a(t)s(t)}{n(t)} + \frac{\psi_2 b(t)s(t)}{n(t)} + \frac{\psi_3 m(t)s(t)}{n(t)} \right) p - (\vartheta + \vartheta_1 + \rho)a(t) \right\} dt + \eta_2 a(t)d\mathcal{B}_2(t),$$

$$db(t) = \left\{ (1-p)\left( \frac{\psi_1 s(t)a(t)}{n(t)} + \frac{\psi_2 b(t)s(t)}{n(t)} + \frac{\psi_3 s(t)m(t)}{n(t)} \right) + q\rho a(t) \right.$$

$$\left. - (\zeta + \vartheta + \vartheta_2)b(t) \right\} dt + \eta_3 b(t)d\mathcal{B}_3(t),$$

$$dr(t) = \{ vs(t) + (1-q)\rho a(t) + \zeta b(t) - \vartheta r(t) \} dt + \eta_4 r(t)d\mathcal{B}_4(t),$$

$$dm(t) = \{ \alpha_2 b(t) + \alpha_1 a(t) - \eta m(t) \} dt + \eta_5 m(t)d\mathcal{B}_5(t),$$

$$(1)$$

where $\Phi$ is the birth rate, $\psi_i$, $i = 1, 2, 3$, are the disease transmission rates. The natural death is assumed to be $\vartheta$ while the moving rate of pre-symptomatic individuals to the symptomatic population is denoted by $\rho$. We also assume that $\vartheta_i$, $i = 1, 2$, are the death rates that occur from disease while $\zeta$ represents the recovery rate of the symptomatic population. Moreover, $\alpha_i$, $i = 1, 2$, are the contributing rates for reservoir, and $\eta_i$, $i = 1, \ldots, 5$, are the intensities of white noise.

### 3.1 WELL-POSEDNESS

This section is devoted to discussing the well-posedness of the proposed epidemic problem. We show that the epidemic model (1) is mathematically as well as biologically feasible by proving that the model has a unique and positive solution. We use the Lyapunov theory and Itô formula to

perform the existence analysis of Eq. (1) which describes by the following theorem. Due to space limit, all the proofs are deferred to appendix.

**Theorem 1** *For any initial population sizes $(s(0), a(0), b(0), r(0), m(0)) \in \mathcal{R}_+^5$, there exists a unique global solution of the model (1) remains in $\mathcal{R}_+^5$, almost surely (a.s).*

**Theorem 2** *For any $(s(0), a(0), b(0), r(0), m(0)) \in \mathcal{R}_+^5$, the solution of model (1) is positive whenever exists.*

### 3.2 EXTINCTION

We discuss the novel disease extinction and find enough conditions for disease extinction in the form of an expression containing the white noise intensities and epidemic parameters. Let us assume that $\mathcal{R}_0$ is the stochastic reproductive parameter defined as $\mathcal{R}_0 = \mathcal{R}_1^E + \mathcal{R}_2^E$, where

$$\mathcal{R}_1^E = \frac{\Phi(\psi_1 + \psi_1 + \psi_1)p}{(\upsilon + \vartheta)\left(\rho + \vartheta + \vartheta_1 + \frac{1}{2}\eta_2^2\right)}, \text{ and } \quad \mathcal{R}_2^E = \frac{\Phi(\psi_1 + \psi_1 + \psi_1)(1 - p)}{(\upsilon + \vartheta)\left(\zeta + \vartheta + \vartheta_2 + \frac{1}{2}\eta_3^2\right)}. \quad (2)$$

Since the reproductive parameter is defined to be the average number of the secondary infectious produced by a single infective individual whenever introduced into a susceptible population, the infection dies out if $\mathcal{R}_0 < 1$, and spreads whenever $\mathcal{R}_0 > 1$. This shows that the threshold parameter $\mathcal{R}_0$ is an important quantity, of which the sensitivity analysis against some chosen parameters is described in the following figures.

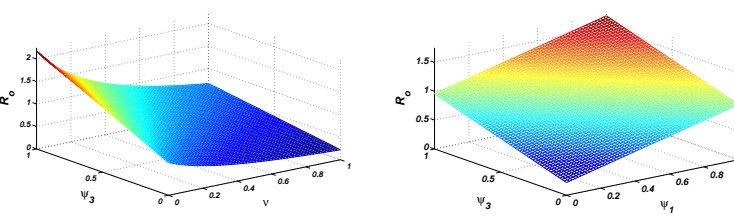

(a) Variation of $\mathcal{R}_0$ verses $\nu$ and $\psi_3$.     (b) Variation of $\mathcal{R}_0$ verses $\psi_1$ and $\psi_2$.

Figure 2: The graphical results show the variation of threshold parameter $\mathcal{R}_0$ against some chosen parameters $\{\psi_1, \psi_3, \nu\}$, while the other parametric values are: $\Phi = 0.9$, $\psi_1 = 0.2$, $\psi_2 = 0.3$, $\nu = 0.53$, $\theta = 0.44$, $\rho = 0.55$, $\theta_1 = 0.45$, $\theta_2 = 0.65$, $p = 0.41$, $\eta_2 = 0.51$ and $\eta_3^2 = 0.65$.

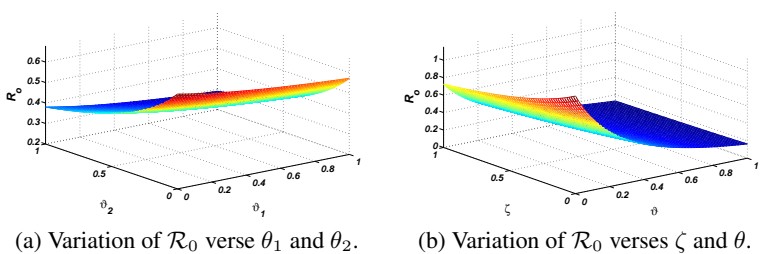

(a) Variation of $\mathcal{R}_0$ verse $\theta_1$ and $\theta_2$.     (b) Variation of $\mathcal{R}_0$ verses $\zeta$ and $\theta$.

Figure 3: This visualizes the variation of threshold parameter $\mathcal{R}_0$ versus some sensitive epidemic parameters $\{\theta_1, \theta_2, \zeta\}$, while the other parametric values are: $\Phi = 0.9$, $\psi_1 = 0.2$, $\psi_2 = 0.3$, $\nu = 0.53$, $\theta = 0.44$, $\rho = 0.55$, $\theta_1 = 0.45$, $\theta_2 = 0.65$, $p = 0.41$, $\eta_2 = 0.51$ and $\eta_3^2 = 0.65$.

Further, regarding disease extinction, the following result is given.

**Theorem 3** *The novel disease of coronavirus dies out, if the stochastic reproductive number is less than one ($\mathcal{R}_0 < 1$) and then*

$$\limsup_{t \to \infty} \frac{\ln a(t)}{t} \leq \left(\rho + \vartheta + \vartheta_1 + \frac{1}{2}\eta_2^2\right)(\mathcal{R}_1^E - 1) < 0,$$

*and*

$$\lim_{t\to\infty} \sup \frac{\ln b(t)}{t} \le \left(\zeta + \vartheta + \vartheta_2 + \frac{1}{2}\eta_3^2\right)(\mathcal{R}_2^E - 1) < 0.$$

*Also*

$$\lim_{t\to\infty} s(t) = \frac{\Phi}{\upsilon + \vartheta}, \;\; \lim_{t\to\infty} r(t) = \frac{\upsilon\Phi}{\vartheta(\vartheta + \upsilon)}, \;\; \lim_{t\to\infty} m(t) = \lim_{t\to\infty} b(t) = \lim_{t\to\infty} a(t) = 0, \; a.s.$$

## 4 MODEL DISCRETIZATION

To perform the numerical assessment of the model that is under consideration, we use Euler–Maruyama (EM) method and Milstein's Higher Order Method. We assume that the desired time interval is $[0, T]$. The time-step $\Delta t = \frac{T}{L}$, where $L$ is a positive integer. A point of the discretized interval is $\tau_j = j\Delta t$. For the sake of simplicity, a solution of the problem $(s(\tau_j), a(\tau_j), b(\tau_j), r(\tau_j), m(\tau_j))$ can be written as $(s_j, a_j, b_j, r_j, m_j)$ and $\mathcal{B}_i(\tau_j) = \mathcal{B}_{i_j}$, $i = 1, 2, \ldots, 5$. The Euler–Maruyama (EM) method for the proposed model will take the form:

$$s_j = s_{j-1} + \left\{\Phi - \frac{\psi_1 a_{j-1} s_{j-1}}{n_{j-1}} - \frac{\psi_2 b_{j-1} s_{j-1}}{n_{j-1}} - \frac{\psi_3 m_{j-1} s_{j-1}}{n_{j-1}} - (\upsilon + \theta)s_{j-1}\right\}\Delta t$$
$$+ \eta_1 s_{j-1}(\mathcal{B}_{1_j} - \mathcal{B}_{1_{j-1}}),$$

$$a_j = a_{j-1} + \left\{p\left(\frac{\psi_1 a_{j-1} s_{j-1}}{n_{j-1}} + \frac{\psi_2 b_{j-1} s_{j-1}}{n_{j-1}} + \frac{\psi_3 m_{j-1} s_{j-1}}{n_{j-1}}\right) - (\rho + \theta + \theta_1)a_{j-1}\right\}\Delta t$$
$$+ \eta_2 a_{j-1}(\mathcal{B}_{2_j} - \mathcal{B}_{2_{j-1}}),$$

$$b_j = b_{j-1} + \left\{(1-p)\left(\frac{\psi_1 a_{j-1} s_{j-1}}{n_{j-1}} + \frac{\psi_2 b_{j-1} s_{j-1}}{n_{j-1}} + \frac{\psi_3 m_{j-1} s_{j-1}}{n_{j-1}}\right) + q\rho a_{j-1}\right. \tag{3}$$
$$\left. - (\zeta + \theta + \theta_2)b_{j-1}\right\}dt + \eta_3 b_{j-1}(\mathcal{B}_{3_j} - \mathcal{B}_{3_{j-1}}),$$

$$r_j = r_{j-1} + \left\{\upsilon s_{j-1} + (1-q)\rho a_{j-1} + \zeta b_{j-1} - \theta r_{j-1}\right\}dt + \eta_4 r_{j-1}(\mathcal{B}_{4_j} - \mathcal{B}_{4_{j-1}}),$$

$$m_j = m_{j-1} + \left\{\alpha_1 a_{j-1} + \alpha_2 b_{j-1} - \eta m_{j-1}\right\}dt + \eta_5 m_{j-1}(\mathcal{B}_{5_j} - \mathcal{B}_{5_{j-1}}).$$

It can be noticed that in the deterministic case ($\eta_i = 0$), Eq. (3) reduces to Euler's method. The next step is to compute discretized Brownian paths that can be used to find $\mathcal{B}_i(\tau_j) - \mathcal{B}_i(\tau_{j-1})$ needed in Eq. (1). For this, let us assume that the step size $\Delta t$ is an integral multiple of $R \ge 1$ of the increment $\delta t$ for the Brownian path. This ensures that the set of points $t'_j s$ (the discretized Brownian path) also contains $\tau_j$ (used in the EM solution). The increments $\mathcal{B}_i(\tau_j) - \mathcal{B}_i(\tau_{j-1})$ in the EM method (3) are given by

$$\mathcal{B}_i(\tau_j) - \mathcal{B}_i(\tau_{j-1}) = \mathcal{B}_i(jR\delta t) - \mathcal{B}_i((j-1)R\delta t) = \sum_{k=j\mathcal{R}-\mathcal{R}+1}^{j\mathcal{R}} d\mathcal{B}_k.$$

If one replaces these Brownian increments by $\sqrt{\Delta t}V_j$ ($V_j$ takes the values $+1$ and $-1$ with equal probability), it will give the weak Euler Maruyama (WEM) procedure. The above steps can be concluded in Algorithm 1. Next, we truncate the Itô–Taylor expansion at a suitable point, which gives Milstein's procedure for SDE (1) as follows:

$$s_j = s_{j-1} + \left\{\Phi - \frac{\psi_1 a_{j-1} s_{j-1}}{n_{j-1}} - \frac{\psi_2 b_{j-1} s_{j-1}}{n_{j-1}} - \frac{\psi_3 m_{j-1} s_{j-1}}{n_{j-1}} - (\upsilon + \theta)s_{j-1}\right\}\Delta t$$
$$+ \eta_1 s_{j-1}(\mathcal{B}_{1_j} - \mathcal{B}_{1_{j-1}}) + \frac{1}{2}\eta_1^2 s_{j-1}((\mathcal{B}_{1_j} - \mathcal{B}_{1_{j-1}})^2 - \Delta t),$$

$$a_j = a_{j-1} + \left\{p\left(\frac{\psi_1 a_{j-1} s_{j-1}}{n_{j-1}} + \frac{\psi_2 b_{j-1} s_{j-1}}{n_{j-1}} + \frac{\psi_3 m_{j-1} s_{j-1}}{n_{j-1}}\right) - (\rho + \theta + \theta_1)a_{j-1}\right\}\Delta t$$
$$+ \eta_2 a_{j-1}(\mathcal{B}_{2_j} - \mathcal{B}_{2_{j-1}}) + \frac{1}{2}\eta_2^2 a_{j-1}((\mathcal{B}_{2_j} - \mathcal{B}_{2_{j-1}})^2 - \Delta t),$$

$$
\begin{aligned}
b_j = b_{j-1} + &\left\{ (1-p)\left( \frac{\psi_1 a_{j-1} s_{j-1}}{n_{j-1}} + \frac{\psi_2 b_{j-1} s_{j-1}}{n_{j-1}} + \frac{\psi_3 m_{j-1} s_{j-1}}{n_{j-1}} \right) + q\rho a_{j-1} \right. \\
&\left. - (\zeta + \theta + \theta_2) b_{j-1} \right\} dt + \eta_3 b_{j-1}(\mathcal{B}_{3_j} - \mathcal{B}_{3_{j-1}}) + \frac{1}{2}\eta_3^2 b_{j-1}((\mathcal{B}_{3_j} - \mathcal{B}_{3_{j-1}})^2 - \Delta t), \\
r_j = r_{j-1} + &\left\{ v s_{j-1} + (1-q)\rho a_{j-1} + \zeta b_{j-1} - \theta r_{j-1} \right\} \Delta t + \eta_4 r_{j-1}(\mathcal{B}_{4_j} - \mathcal{B}_{4_{j-1}}) \\
&+ \frac{1}{2}\eta_4^2 r_{j-1}((\mathcal{B}_{4_j} - \mathcal{B}_{4_{j-1}})^2 - \Delta t), \\
m_j = m_{j-1} + &\left\{ \alpha_1 a_{j-1} + \alpha_2 b_{j-1} - \eta m_{j-1} \right\} \Delta t + \eta_5 m_{j-1}(\mathcal{B}_{5_j} - \mathcal{B}_{5_{j-1}}) \\
&+ \frac{1}{2}\eta_5^2 m_{j-1}((\mathcal{B}_{5_j} - \mathcal{B}_{5_{j-1}})^2 - \Delta t).
\end{aligned}
\tag{4}
$$

The main algorithm of Milstein's Higher Order Method for the model is given in Algorithm 2.

## 5  EXPERIMENTAL RESULTS

In this section, we visualize the large-scale numerical simulations of our model as described by Eq. (1). In the case of epidemiological models, it is very important to validate and justify the model with the help of available data. We fit our model to the data of SARS-CoV-2 reported in the United Arab Emirates (UAE) between March 21st, 2022 - Jun 21st, 2022, which clearly shows the effectiveness of the model and the importance of the current study as shown in Figure 4. To fit the model, parameters need be estimated. First, we import the data and accordingly use the theory of the Runge-Kutta method for the numerical solution of the proposed system. We then use the nonlinear fitting function 'lsqcurvefit' in matlab to estimate the epidemic parameter as presented in table 1. The error analysis of the model predictions with real data is also performed by calculating the error and absolute error as presented in Figure 4. Both the error and absolute error describe that within the first 60 days the error curve fluctuates in the neighborhood of 0. That is the model best fitting the data. Within this duration, one can say that the model covers above 95% of the data points. Between 65 and 85 days, there is a gap between the observed data and the curve predicted by the model. This is because, we have a sudden increase in the reported cases and mathematically, such a drastic change in the dynamics of infection is very difficult to capture. Afterward, the error tends to decrease, and the model again best fits the data which shows the effectiveness of the present work. We use the schemes as reported in Algorithms 1-2 with an application of model parameters

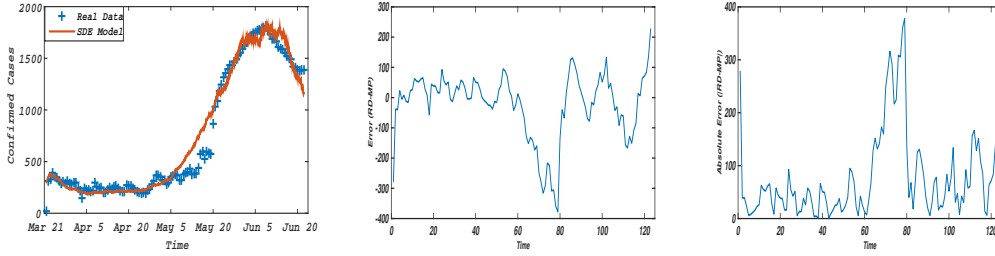

Figure 4: Parameter estimation and error analysis for the proposed model based on the data from United Arab Emirates (UAE). The model fitted with the real data of daily new confirmed cases for the time period March 21, 2022 to June 21, 2022. Moreover, calculated the error analysis of the values via simulation and real data.

in a biologically feasible way, and initial sizes of populations, i.e., $(20, 30, 35, 32, 50)$. We also assume that the unit of the time interval is 0 to 10. Thus, the execution of the schemes as reported above with the biologically feasible parametric value and initial compartmental population sizes lead to the results as depicted in Figures 5, 6, and 7, where the solution curves of the proposed epidemic problem (1) are represented by Figure 5. Particularly, the numerical solution retrieved by

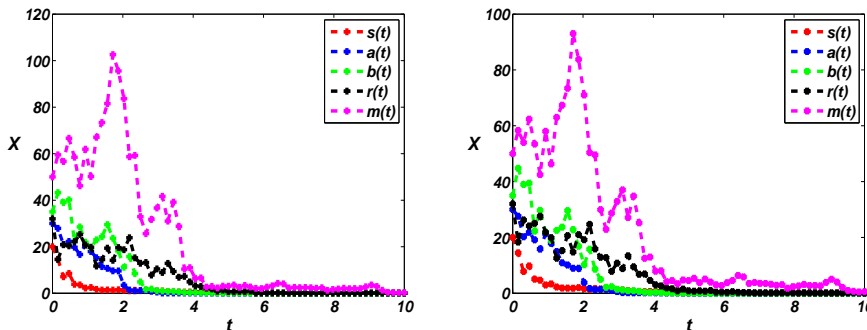

(a) Solution curves via Euler–Maruyama (EM) method.
(b) Solution curves via Milstein's method.

Figure 5: The plots visualizes the solution curves of the model (1) against the parameter values: $\Phi = 0.1$, $\psi_1 = 0.2$, $\psi_3 = 0.01$, $\psi_2 = 0.3$, $\nu = 0.43$, $\theta = 0.44$, $\eta_1 = 0.41$, $\eta_2 = 0.51$, $\eta_3 = 0.65$, $\eta_4 = 0.731$, $\eta_5 = 0.64$, $\eta = 0.4$, $p = 0.41$, $\rho = 0.55$, $\theta_1 = 0.45$, $q = 0.22$, $\zeta = 0.01$, $\theta_2 = 0.65$, $\alpha_1 = 0.43$ and $\alpha_2 = 0.87$.

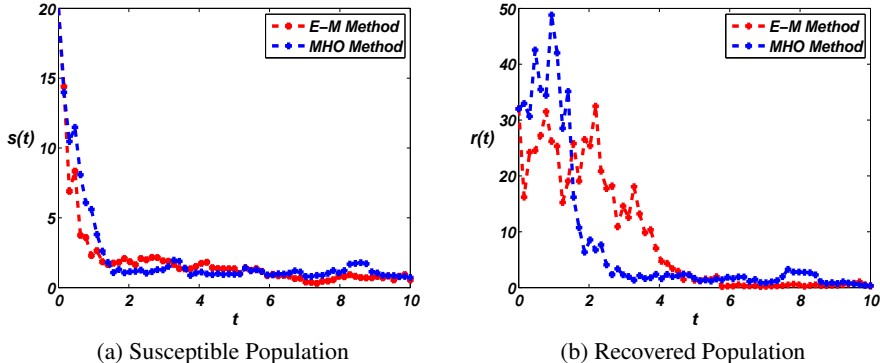

(a) Susceptible Population
(b) Recovered Population

Figure 6: The graphs represent the temporal dynamics of the compartmental population as well as the comparative analysis of the results against parametric values: $\Phi = 0.9$, $\psi_1 = 0.2$, $\psi_3 = 0.01$, $\psi_2 = 0.3$, $\nu = 0.43$, $\theta = 0.44$, $\eta_1 = 0.41$, $\eta_2 = 0.51$, $\eta_3 = 0.65$, $\eta_4 = 0.731$, $\eta_5 = 0.64$, $\eta = 0.6$, $p = 0.41$, $\rho = 0.65$, $\theta_1 = 0.45$, $q = 0.022$, $\zeta = 0.1$, $\theta_2 = 0.65$, $\alpha_1 = 0.43$ and $\alpha_2 = 0.87$.

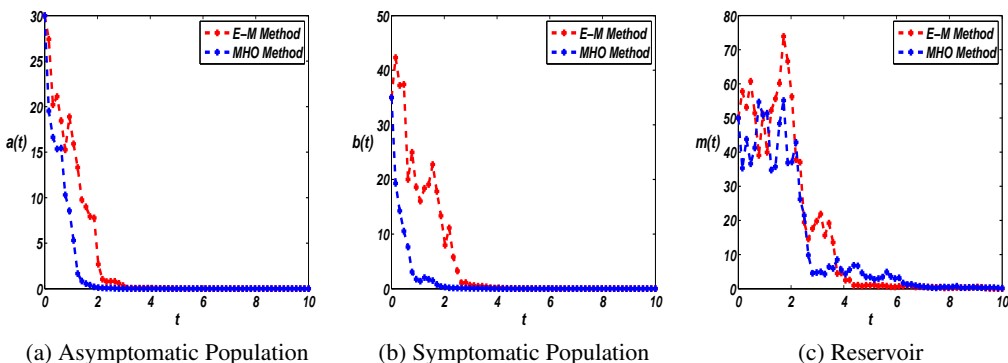

(a) Asymptomatic Population
(b) Symptomatic Population
(c) Reservoir

Figure 7: The graphs represent the dynamics of the compartmental population as well as the comparative analysis of the results against parametric values: $\Phi = 0.9$, $\psi_1 = 0.2$, $\psi_3 = 0.01$, $\psi_2 = 0.3$, $\nu = 0.43$, $\theta = 0.44$, $\eta_1 = 0.41$, $\eta_2 = 0.51$, $\eta_3 = 0.65$, $\eta_4 = 0.731$, $\eta_5 = 0.64$, $\eta = 0.6$, $p = 0.41$, $\rho = 0.65$, $\theta_1 = 0.45$, $q = 0.022$, $\zeta = 0.1$, $\theta_2 = 0.65$, $\alpha_1 = 0.43$ and $\alpha_2 = 0.87$.

Table 1: The values of epidemic parameters of model (1) estimated from real data of SARS-CoV-2

| Parameters | Description | Value | Source |
|---|---|---|---|
| $\psi_1$ | Disease transmission rate due to pre-symptomatic | 0.1387 | Fitted |
| $\psi_2$ | Disease transmission co-efficient due to symptomatic | 0.6174 | Fitted |
| $\psi_3$ | Disease transmission rate from reservoir | 0.001024 | Fitted |
| $v$ | Vaccination | 0.00001 | Fitted |
| $p$ | Probability at which the susceptible become pre-symptomatic | 0.0003 | Fitted |
| $q$ | Probability at which pre-symptomatic become recover | 0.0102 | Fitted |
| $\rho$ | Recovery rate of pre-symptomatic | 0.09 | Fitted |
| $\vartheta_1$ | Death rate due to disease | 0.0054 | Fitted |
| $\vartheta_2$ | Death rate due to disease | 0.5245 | Fitted |
| $\alpha_1$ | Contributing rate for reservoir | 0.3436 | Fitted |
| $\alpha_2$ | Contributing rate for reservoir | 0.0083 | Fitted |
| $\zeta$ | Recovery rate of symptomatic individuals | 0.001 | Fitted |
| $\eta$ | Removal rate of reservoir | 0.01 | Fitted |

Euler–Maruyama (EM) is visualized in Figure 5a while the graphs depicted in Figure 5b show the solution trajectories carried out with the help of Milstein's method.

Further, to show the comparison of solution for the compartmental populations of model (1), we present the temporal dynamics of susceptible, asymptomatic, symptomatic, recovered and reservoir in Figure 6a, 6b, 7a, 7b and 7c, respectively. More precisely, the solution curves of the model (1) for the susceptible population are demonstrated in Figure 6a. The two trajectories represented by the red and blue dashed lines, respectively, show the solution $s(t)$ applying Euler–Maruyama (EM) and Milstein's methods as in Figure 6a. Similarly, the long-run numerical simulation for the pre-asymptomatic, symptomatic, recovered, and reservoir visualized by the blue dashed line and red dashed line, respectively, represents the solution retrieved by Euler–Maruyama (EM) and Milstein's methods, see Figure 6b-7c. It could be observed that the numerical solutions of the proposed problem via Euler–Maruyama (EM) and Milstein's methods agree with each other in the long run, which shows the significance of the developed algorithm. Likewise, the numerical verification of the theoretical results as stated by Theorem 3 is shown in Figure 6 and 7. We obtained this by calculating the value of threshold parameter $\mathcal{R}_0 = 0.2033 < 1$ in this case. The numerical results verify the analytical findings that whenever $\mathcal{R}_0 < 1$ the infection of novel disease dies out (infection population vanishes) while there will be only susceptible and recovered population as shown in Figure 6. Thus, it could be concluded that the elimination of the disease continuously depends on the threshold parameter, and it is very necessary to keep its value less than unity as much as possible.

## 6 CONCLUSION

This paper formulated a stochastic epidemiological model for SARS-CoV-2 virus mitigation according to the characteristics of newly reported coronavirus transmission. The impact of pre-symptomatic, symptomatic, and reservoir has been incorporated to present accurate and well-structured dynamics behavior of the disease. For this purpose, we divided the total compartmental population into the various classes of susceptible, pre-symptomatic, symptomatic, recovered, and reservoir. We showed the proper disease mitigation with the help of the stochastic differential equations model. We studied the existence and uniqueness analysis of the model and showed that the model is biologically as well as mathematically feasible. We also discussed the extinction of the model and calculated some sufficient conditions for the novel disease extinction. To perform the long-run simulation of the proposed model Euler-Maruyama and Milstein's schemes have been developed and supported the analytical findings with graphical representation. We also used real data of the SARS-CoV-2 virus and fitted the model to the data and proved the effectiveness of the current study. In the future, we will work on the dynamics and control of novel corona virus disease using multiple strained model.

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

APPENDIX

A.1 PROOF OF THEOREM 1

**Proof.** To begin, we use the methodology adopted in Lei & Yang (2017) which gives that for any initial size of population $(s(0), a(0), b(0), r(0), m(0)) \in \mathcal{R}_+^5$, the proposed epidemic system (1) admits a unique local solution $(s(t), a(t), b(t), r(t), m(t)$ on $t \in [0, \tau_e)$ (where $\tau_e$ is the explosion time). We investigate that $\tau_e = \infty$ to prove that the solution is global *a.s.* To do, we assume that $\kappa_0 \geq 0$ is sufficiently large such that $\frac{1}{\kappa_0} < (s(0), a(0), b(0), r(0), m(0)) < \kappa_0$. We define the stopping time, $\tau_k$ for any $\kappa \geq \kappa_0$ by

$$\tau_k = \inf \left\{ t \in [0, \tau_e) : \min\{s(t), a(t), b(t), r(t), m(t)\} \leq \frac{1}{k} \quad \text{or} \right.$$

$$\left. \max\{s(t), a(t), b(t), r(t), m(t)\} \geq \kappa \right\}.$$

Let us assume that $\phi$ is the null set and $\inf \phi = \infty$. Obviously, $\tau_k$ increasing whenever $k$ increasing unboundedly. We also set $\lim \tau_\kappa = \tau_\infty$ whenever $k \to \infty$, then $\tau_e \geq \tau_\infty$. If we can show that $\tau_\infty = \infty$ then $\tau_e = \infty$ will also hold. To complete the proof, only we need to show that $\tau_e = \infty$. So, we are going to prove it on contrary basis that is if this statement is false, there exists a pair of constants $T > 0$ and $\varepsilon \in (0, 1)$, such that

$$P\{\tau_\kappa \leq T\} \geq \varepsilon, \tag{5}$$

for each $\kappa \geq \kappa_0$. Now define a function $\mathcal{H} \in \mathcal{C}^2$ by $\mathcal{H} : \mathcal{R}_+^5 \to \mathcal{R}_+$ by

$$\mathcal{H}(s, a, b, r, m) = s - 1 - \ln s + a - 1 - \ln a + b - 1 - \ln b + r - 1 - \ln r + m - 1 - \ln m.$$

Clearly, $\mathcal{H}$ is non-negative, and thus by the virtue of the Itô formula, we obtain

$$d\mathcal{H} = \left(1 - \frac{1}{s}\right) ds + \left(1 - \frac{1}{a}\right) da + \left(1 - \frac{1}{b}\right) db + \left(1 - \frac{1}{r}\right) dr + \left(1 - \frac{1}{m}\right) dm$$

$$+ \frac{1}{2} \left(\eta_1^2 + \ldots + \eta_5^2\right) dt + \eta_1 (1 - s) d\mathcal{B}_1(t) + \eta_2 (1 - a) d\mathcal{B}_2(t) \tag{6}$$

$$+ \eta_3 (1 - b) d\mathcal{B}_3(t) + \eta_4 (1 - r) d\mathcal{B}_4(t) + \eta_5 (1 - m) d\mathcal{B}_5(t).$$

Since $s + a + b + r + m = n$, then using Eq.(1) in Eq.(6) with a little simplification and re-arrangement implies

$$d\mathcal{H} \leq (\Phi + 2(\psi_1 + \psi_2 + \psi_3) + \upsilon + \vartheta + \rho + \zeta + 4\vartheta + \vartheta_1 + \vartheta_2 + \eta + \alpha_1 + \alpha_2 + \eta) dt$$

$$+ \frac{1}{2} \left(\eta_1^2 + \ldots + \eta_5^2\right) dt - \eta_1 (s - 1) d\mathcal{B}_1(t) - \eta_2 (a - 1) d\mathcal{B}_2(t)$$

$$- \eta_3 (b - 1) d\mathcal{B}_3(t) - \eta_4 (r - 1) d\mathcal{B}_4(t) - \eta_5 (m - 1) d\mathcal{B}_5(t).$$

For the shake of simplicity, let us assume that $\mathcal{K} = \Phi + 2(\psi_1 + \psi_2 + \psi_3) + \upsilon + \vartheta + \rho + \zeta + 4\vartheta + \vartheta_1 + \vartheta_2 + \eta + \alpha_1 + \alpha_2 + \eta + \frac{1}{2} \left(\eta_1^2 + \ldots + \eta_5^2\right)$, the above inequality looks like

$$d\mathcal{H} \leq \mathcal{K}dt - \eta_1 (s - 1) d\mathcal{B}_1(t) - \eta_2 (a - 1) d\mathcal{B}_2(t)$$

$$- \eta_3 (b - 1) d\mathcal{B}_3(t) - \eta_4 (r - 1) d\mathcal{B}_4(t) - \eta_5 (m - 1) d\mathcal{B}_5(t).$$

For any $t \in [0, T]$ and $\kappa \geq \kappa_0$, the application of integration with limits 0 to $\tau_k \wedge t$ gives

$$\int_0^{\tau_k \wedge t} d\mathcal{H} \leq \int_0^{\tau_k \wedge t} \mathcal{K}dt - \int_0^{\tau_k \wedge t} (s - 1)\eta_1 d\mathcal{B}_1 - \int_0^{\tau_k \wedge t} (a - 1)\eta_2 d\mathcal{B}_2$$

$$- \int_0^{\tau_k \wedge t} (b - 1)\eta_3 d\mathcal{B}_3 - \int_0^{\tau_k \wedge t} (r - 1)\eta_4 d\mathcal{B}_3 - \int_0^{\tau_k \wedge t} (m - 1)\eta_5 d\mathcal{B}_5.$$

By taking the expectation we may arrive at

$$\mathcal{E} \left\{ \mathcal{H}(s(\tau_k \wedge t), a(\tau_k \wedge t), b(\tau_k \wedge t), r(\tau_k \wedge t), m(\tau_k \wedge t)) \right\}$$

$$\leq \mathcal{H}(s(0), a(0), b(0), r(0), m(0)) + \mathcal{E} \left\{ \int_0^t \mathcal{K}dv \right\},$$

which implies

$$\mathcal{E}\left\{\mathcal{H}(s(\tau_k \wedge t), a(\tau_k \wedge t), b(\tau_k \wedge t), r(\tau_k \wedge t), m(\tau_k \wedge t))\right\}$$
$$\leq \mathcal{H}(s(0), a(0), b(0), r(0), m(0)) + T\mathcal{K}. \tag{7}$$

Let $\Delta_\kappa = \{T \geq \tau_\kappa\}$ for every $\kappa \geq \kappa_0$, then $P(\Delta_\kappa) \geq \varepsilon$. Note that some components of $s(\tau_k \wedge T)$, $a(\tau_k \wedge T)$, $b(\tau_k \wedge T)$, $r(\tau_k \wedge T)$, $m(\tau_k \wedge T)$ equal either $1/\kappa$ or $\kappa$, and $\mathcal{H}(s(\tau_k \wedge t), a(\tau_k \wedge t), b(\tau_k \wedge t), r(\tau_k \wedge t), m(\tau_k \wedge t))$ is not less than $\frac{1}{\kappa} + \ln \kappa - 1$ or $-\ln \kappa + \kappa - 1$ for all $\varpi \in \Delta_\kappa$, thus we have

$$\mathcal{H}(s, a, b, c, m) \geq \left(-\log \kappa - 1 + \kappa\right) \wedge \left(\frac{1}{\kappa} - 1 + \log \kappa\right).$$

So from Eq.(5) and (7) we may conclude that

$$\mathcal{H}((s(0), a(0), b(0), r(0), m(0))) + T\mathcal{K}$$
$$\geq \mathcal{E}\left\{1_{\Delta\kappa(\varpi)}\mathcal{H}\left(s(\tau_\kappa \wedge T), a(\tau_\kappa \wedge T), b(\tau_\kappa \wedge T), r(\tau_\kappa \wedge T), m(\tau_\kappa \wedge T)\right)\right\},$$
$$\geq \mathcal{E}\left\{1_{\Delta\kappa(\varpi)}\left(\log \kappa - 1 + \frac{1}{\kappa}\right) \wedge (-\log \kappa - 1 + \kappa)\right\},$$
$$= \left\{\log \kappa - 1 + \frac{1}{\kappa}\right\} \wedge (-\log \kappa - 1 + \kappa)E\left(1_{\Delta\kappa(\varpi)}\right).$$

The last inequality gives

$$\mathcal{H}((s(0), a(0), b(0), r(0), m(0))) + T\mathcal{K} \geq \varepsilon\left(\ln \kappa + \frac{1}{\kappa} - 1\right) \wedge (-\log \kappa + \kappa - 1),$$

where $1_{\Delta\kappa(\varpi)}$ is the indicator function for $\Delta_\kappa(\varpi)$. Thus for $k$ approaches $\infty$ we may derive that $\infty = \mathcal{H}\left((s(0), a(0), b(0), r(0), m(0))\right) + \mathcal{K}T < \infty$, which is a contradiction, so $\tau_e = \infty$ a.s.

## A.2 PROOF OF THEOREM 2

**Proof.** Let $\mathcal{L} \in [0, \infty)$ and we suppose that the solution of the model (1) exists in $\mathcal{L}$, then for every $t \in \mathcal{L}$, we have

$$s(t) = \exp\left\{-(\upsilon + \vartheta)t - \int_0^t \left(\frac{\psi_1 a + \psi_2 b + \psi_3 m}{n} + \frac{\eta_1^2}{2}\right) du\right\} - \eta_1 \int_0^t d\mathcal{B}_1(u)$$
$$\times \left[s(0) + \Phi \int_0^t \exp\left\{(\varepsilon + \vartheta)t + \int_0^u \left(\frac{\psi_1 a + \psi_2 b + \psi_3 m}{n} + \frac{\eta_1^2}{2}\right) dv + \eta_1 \int_0^u d\mathcal{B}_1(v)\right\} du\right].$$

Next, the solution of the second equation of the proposed model looks like

$$a(t) = \exp\left\{-(\rho + \vartheta + \vartheta_1)t + \int_0^t \left(\frac{p\psi_1 s}{n} + \frac{\eta_2^2}{2}\right) du\right\} + \eta_2 \int_0^t d\mathcal{B}_2(u) \times \left[a(0)\right.$$
$$\left. + \int_0^t p\left(\frac{\psi_1 bs + \psi_3 ms}{n}\right) \exp\left\{(\rho + \vartheta + \vartheta_1)t - \int_0^s \left(\frac{p\psi_1 s}{n} + \frac{\eta_2^2}{2}\right) du + \eta_2 \int_0^s d\mathcal{B}_2(u)\right\} ds\right].$$

Clearly, the above equation implies that $s > 0$ and $a \geq 0$. In a similar fashion it can shown that $b$, $r$ and $m$ are non-negative.

A.3 PROOF OF THEOREM 3

**Proof**. To begin, first we integrate the proposed model (1) which yields

$$\int_0^t ds(y) = \Phi t - \int_0^t \left\{ \frac{\psi_1 as}{n} + \frac{\psi_2 bs}{n} + \frac{\psi_3 ms}{n} - (\upsilon + \vartheta)s \right\} dy + \int_0^t \eta_1 s(x) d\mathcal{B}_1(y),$$

$$\int_0^t da(y) = \int_0^t p \left\{ \frac{\psi_1 as}{n} + \frac{\psi_2 bs}{n} + \frac{\psi_3 ms}{n} - (\rho + \vartheta + \vartheta_1) a \right\} dy + \int_0^t \eta_2 a(y) d\mathcal{B}_2(y),$$

$$\int_0^t db(y) = \int_0^t (1-p) \left\{ \frac{\psi_1 as}{n} + \frac{\psi_2 bs}{n} + \frac{\psi_3 ms}{n} - (\zeta + \vartheta + \vartheta_2) a + q\rho a \right\} dy$$

$$+ \int_0^t \eta_2 a(y) d\mathcal{B}_2(y),$$

$$\int_0^t dr(y) = \int_0^t (\upsilon s(x) + (1-q)\rho a + \zeta b - \vartheta r(y)) dy + \int_0^t \eta_4 r(y) d\mathcal{B}_4(y),$$

$$\int_0^t dm(y) = \int_0^t (\alpha_2 b(x) + \alpha_1 a(x) - \eta m(y)) dy + \int_0^t \eta_5 m(y) d\mathcal{B}_5(y),$$

implies that

$$\frac{s(t) - s(0)}{t} = \Phi - \psi_1 \left\langle \frac{as}{n} \right\rangle + \left\langle \frac{\psi_2 bs}{n} \right\rangle + \left\langle \frac{\psi_3 ms}{n} \right\rangle - (\upsilon + \vartheta) \langle s \rangle + \frac{1}{t} \int_0^t \eta_1 s(y) d\mathcal{B}_1(y),$$

$$\frac{a(t) - a(0)}{t} = p \left\{ \psi_1 \left\langle \frac{as}{n} \right\rangle + \psi_2 \left\langle \frac{bs}{n} \right\rangle + \psi_3 \left\langle \frac{ms}{n} \right\rangle - (\rho + \vartheta + \vartheta_1) \langle a \rangle \right\} + \frac{1}{t} \int_0^t \eta_2 a(y) d\mathcal{B}_2(y),$$

$$\frac{b(t) - b(0)}{t} = (1-p) \left\{ \psi_1 \left\langle \frac{as}{n} \right\rangle + \psi_2 \left\langle \frac{bs}{n} \right\rangle + \psi_3 \left\langle \frac{ms}{n} \right\rangle - (\zeta + \vartheta + \vartheta_2) \langle a \rangle + q\rho \langle a \rangle \right\}$$

$$+ \frac{1}{t} \int_0^t \eta_2 a(y) d\mathcal{B}_2(y),$$

$$\frac{r(t) - r(0)}{t} = \upsilon \langle s \rangle + (1-q)\rho \langle a \rangle + \zeta \langle b \rangle - \vartheta \langle r \rangle + \frac{1}{t} \int_0^t \eta_4 r(y) d\mathcal{B}_4(y),$$

$$\frac{m(t) - m(0)}{t} = \alpha_2 \langle b \rangle + \alpha_1 \langle a \rangle - \eta \langle m \rangle + \frac{1}{t} \int_0^t \eta_5 m(y) d\mathcal{B}_5(y).$$

Adding the first three equations we obtain

$$\frac{s(t) + a(t) + b(t) - s(0) - a(0) - b(0)}{t} = \Phi - (\upsilon + \vartheta) \langle s \rangle - (\rho + \vartheta + \vartheta_1) \langle a \rangle - (\zeta + \vartheta + \vartheta_2) \langle b \rangle$$

$$+ \frac{1}{t} \int_0^t \{ \eta_1 s(y) d\mathcal{B}_1(y) + \eta_2 a(y) d\mathcal{B}_2(y) + \eta_3 b(y) d\mathcal{B}_3(y) \}.$$

Using some algebraic manipulation and little re-arrangement we arrive at

$$\langle s(t) \rangle = \frac{\Phi}{\upsilon + \vartheta} - \frac{\rho(1-q)}{\upsilon + \vartheta} \langle a \rangle - (\vartheta + \vartheta_1) \langle a \rangle - (\zeta + \vartheta + \vartheta_2) \langle b \rangle + f(t), \tag{8}$$

where

$$f(t) = -\frac{1}{\upsilon + \vartheta} \left\{ \frac{a(t) + b(t) + s(t) - b(0) + a(0) + s(0)}{t} \right\}$$

$$+ \frac{1}{t} \int_0^t \{ \eta_1 s(y) dB_1(y) + \eta_2 a(y) dB_2(y) + \eta_3 b(y) dB_3(y) \}.$$

Clearly, $f(t)$ tend to 0 if $t$ increasing unboundedly

$$\lim_{t \to \infty} f(t) = 0 \ a.s.$$

Now by Itô formula the second equation of model (1) leads to the following assertion

$$d \ln a(t) = p \left\{ \frac{\psi_1 s}{n} + \frac{\psi_2 bs}{an} + \frac{\psi_3 ms}{an} \right\} - (\rho + \vartheta + \vartheta_1) - \frac{1}{2} \eta_2^2 + \eta_2 dB_2(t).$$

Applying integration we get

$$\frac{1}{t}\ln a(t)_0^t = p\left\{\psi_1\left\langle\frac{s}{n}\right\rangle + \psi_2\left\langle\frac{bs}{an}\right\rangle + \psi_3\left\langle\frac{ms}{an}\right\rangle\right\} - (\rho + \vartheta + \vartheta_1) - \frac{1}{2}\eta_2^2 + \frac{1}{t}\eta_2 d\mathcal{B}_2(t).$$

Since $s + a + b + r + m = n$, so the above equation may takes the following form

$$\frac{1}{t}\ln a(t)_0^t \le p\left\{(\psi_1 + \psi_2 + \psi_3)\langle s\rangle\right\} - (\rho + \vartheta + \vartheta_1) - \frac{1}{2}\eta_2^2 + \frac{1}{t}\eta_2 d\mathcal{B}_2(t). \tag{9}$$

Plugging the value of $\langle s\rangle$ by following Eq.(8) in (9) we reach to the following assertion

$$\frac{1}{t}\ln a(t)_0^t \le p\left(\psi_1 + \psi_2 + \psi_3\right)\left\{\frac{\Phi}{\upsilon + \vartheta} - \frac{\rho(1-q)}{\upsilon + \vartheta}\langle a\rangle - \frac{\vartheta + \vartheta_1}{\upsilon + \vartheta}\langle a\rangle - (\rho + \vartheta + \vartheta_2)\langle b\rangle\right.$$
$$\left. + f(t)\right\} - (\rho + \vartheta + \vartheta_1) - \frac{1}{2}\eta_2^2 + \frac{1}{t}\eta_2 d\mathcal{B}_2(t),$$
$$\le p\left(\psi_1 + \psi_2 + \psi_3\right)\left\{\frac{\Phi}{\upsilon + \vartheta} + f(t)\right\} - (\rho + \vartheta + \vartheta_1) - \frac{1}{2}\eta_2^2 + \frac{1}{t}\eta_2 d\mathcal{B}_2(t).$$

Following *strong law of large number* Birkel (1988) as well as using Eq.(2), the last inequality may takes the form

$$\limsup_{t\to\infty}\frac{\ln a(t)}{t} \le \left\{\rho + \vartheta + \vartheta_1 + \frac{1}{2}\eta_2^2\right\}(\mathcal{R}_1^E - 1) < 0 \ a.s.,$$

implies that $\lim a(t) = 0$ whenever $\mathcal{R}_1^E < 1$. In a similar the third equation of model (1) implies that

$$d\ln b(t) = (1 - p)\left\{\frac{\psi_1 as}{n} + \frac{\psi_2 s}{n} + \frac{\psi_3 ms}{an}\right\} + \frac{q\rho a}{b} - (\zeta + \vartheta + \vartheta_2) - \frac{1}{2}\eta_3^2 + \eta_3 d\mathcal{B}_3(t).$$

Taking the integral of both sides, we derive

$$\frac{1}{t}\ln b(t)_0^t = (1 - p)\left\{\psi_1\left\langle\frac{as}{bn}\right\rangle + \psi_2\left\langle\frac{s}{n}\right\rangle + \psi_3\left\langle\frac{ms}{bn}\right\rangle\right\} + q\rho\left\langle\frac{a}{b}\right\rangle - (\zeta + \vartheta + \vartheta_2)$$
$$- \frac{1}{2}\eta_3^2 + \frac{1}{t}\eta_3 d\mathcal{B}_3(t).$$

Following the similar procedure as above we arrive at

$$\frac{1}{t}\ln b(t)_0^t \le (1 - p)\left(\psi_1 + \psi_2 + \psi_3\right)\left\{\frac{\Phi}{\upsilon + \vartheta} - \frac{\rho(1-q)}{\upsilon + \vartheta}\langle a\rangle - \frac{\vartheta + \vartheta_1}{\upsilon + \vartheta}\langle a\rangle - (\rho + \vartheta + \vartheta_2)\langle b\rangle\right.$$
$$\left. + f(t)\right\} - (\zeta + \vartheta + \vartheta_2) - \frac{1}{2}\eta_3^2 + \frac{1}{t}\eta_3 d\mathcal{B}_2(t),$$
$$\le (1 - p)\left(\psi_1 + \psi_2 + \psi_3\right)\left\{\frac{\Phi}{\upsilon + \vartheta} + f(t)\right\} - (\zeta + \vartheta + \vartheta_2) - \frac{1}{2}\eta_3^2 + \frac{1}{t}\eta_3 d\mathcal{B}_2(t).$$

Substituting Eq.(2), and following again the *strong law of large number*, so the final inequality leads to

$$\limsup_{t\to\infty}\frac{\ln b(t)}{t} \le \left\{\zeta + \vartheta + \vartheta_2 + \frac{1}{2}\eta_3^2\right\}(\mathcal{R}_2^E - 1) < 0,$$

which implies that $\lim b(t) = 0$ for $t$ tend to $\infty$ whenever $\mathcal{R}_2^E < 1$. Moreover, the last equation of model (1) yields that $m(t) = 0$ if $t \to \infty$. Thus, Eq.(8) gives that $\lim s = \frac{\Phi}{\upsilon + \vartheta}$. Therefore, we conclude that the extinction of the novel disease depend continuously on $\mathcal{R}_0$, and so vanishes whenever the value of $\mathcal{R}_0$ is less then unity.

A.4 ALGORITHMS

---

**Algorithm 1** Euler Maruyama Method (EMM)

---

1: **Descretization** of $[0, T]$ into $L$ equal subintervals of width $\Delta t = \frac{T}{L} > 0$: $0 = \tau_0 < \tau_1 < \cdots < \tau_L = T$.
2: **Setting** the initial data $(s_0, a_0, b_0, r_0, m_0)$.
3: **Recursive** definition of $(s_j, a_j, b_j, r_j, m_j)$ for $1 \leq j \leq L$ as in (3).
4: **Discretization** of the Brownian paths that can be used in finding $\mathcal{B}_i(\tau_j) - \mathcal{B}_i(\tau_{j-1})$ using the constant $\mathcal{R} \geq 1$ and the increment $\delta t$ for the Brownian path.
5: **Finding** of $\mathcal{B}_i(\tau_j) - \mathcal{B}_i(\tau_{j-1}) = \mathcal{B}_i(j\mathcal{R}\delta t) - \mathcal{B}_i((j-1)\mathcal{R}\delta t) = \sum_{k=j\mathcal{R}-\mathcal{R}+1}^{j\mathcal{R}} d\mathcal{B}_k$.

---

**Algorithm 2** Milstein's Higher Order Method (MHOM)

---

1: **Descretization** of $[0, T]$ into $L$ equal subintervals of width $\Delta t = \frac{T}{L} > 0$: $0 = \tau_0 < \tau_1 < \cdots < \tau_L = T$ with $\tau_n = n\Delta t$.
2: **Setting** the initial data $(s_0, a_0, b_0, r_0, m_0)$.
3: **Recursive** definition of $(s_j, a_j, b_j, r_j, m_j)$ for $1 \leq j \leq L$ as in (4).
4: **Discretization** of the Brownian paths that can be used in finding $\mathcal{B}_i(\tau_j) - \mathcal{B}_i(\tau_{j-1})$ using the constant $R \geq 1$ and the increment $\delta t$ for the Brownian path.
5: **Finding** of $\mathcal{B}_i(\tau_j) - \mathcal{B}_i(\tau_{j-1}) = \mathcal{B}_i(j\mathcal{R}\delta t) - \mathcal{B}_i((j-1)\mathcal{R}\delta t) = \sum_{k=j\mathcal{R}-\mathcal{R}+1}^{j\mathcal{R}} d\mathcal{B}_k$.

---

