# OpenReview forum: "Enhancement and Numerical Assessment of Novel SARS-CoV-2 Virus Transmission Model"
_ICLR.cc/2023/Conference — Submitted to ICLR 2023_

### Official Review · Reviewer_wtTR · 2022-10-25

**Confidence:** 5
**Clarity, Quality, Novelty And Reproducibility:** Please see detailed comments above.
**Correctness:** 3
**Technical Novelty And Significance:** 3
**Empirical Novelty And Significance:** Not applicable
**Recommendation:** 3

**Strength And Weaknesses:**

This work has a valid motivation, and it is well-organized. Also, it is technically sound with both theoretical support and empirical validations. However, this work can be further improved in the following aspects:

- Fig. 1 is supposed to be a vector graph.
- I have concerns about the impact of this paper. This work studies the method to track and control COVID, which does not use any ML or DL methods. Therefore, it may only bring limited insight or impact to the ML community. On the other hand, though a new compartmental model has been proposed, similar epidemic models have also been developed in recent studies. Meanwhile, this work does not provide any observations or suggestions which are not obvious or counterintuitive, which may also bring limited insights to the policymakers.
- The form of Table 1 should be revised to meet the requirement of the conference.
- It would be better to have some discussions after each theorem.
- In equation 3, commas are missing in the first two equations.


**Summary Of The Paper:**

This work considers the SARS-CoV-2 transmission and forecast issue. To address this problem, a new model based on stochastic differential equations is proposed. For the proposed model, the necessary conditions for disease eradication are developed and also verified theoretically. To further validate the proposed model, numerical simulations based on real data are also provided.

**Summary Of The Review:**

In my opinion, this work may not be suitable to be published in ICLR. I think it can only bring limited insights to both ML and optimization community.

---

### Official Review · Reviewer_qnjX · 2022-11-01

**Confidence:** 2
**Correctness:** 3
**Technical Novelty And Significance:** 3
**Empirical Novelty And Significance:** 2
**Recommendation:** 3

**Clarity, Quality, Novelty And Reproducibility:**

Paper is well-written, and well structured. I do not see the code released for reproducibility.

**Strength And Weaknesses:**

It is a mathematical modelling paper, there is no representation or learning. I don not think that this is within the scope of ICLR, and the audience will have limited interest. As a reviewer I also don't think that I can properly review this paper, given it is from an adjacent field.

**Summary Of The Paper:**

This paper presents a new compartment based epidemic modelling for COVID which considers pre-symptomatic and symptomatic phases. The paper improves a prior model by Khan et al, 2021, and show empirically the new model better fits to a data from UAE.

**Summary Of The Review:**

Paper provides an epidemic modelling of COVID which is not ML related.

---

### Official Review · Reviewer_mAbY · 2022-11-01

**Confidence:** 5
**Correctness:** 3
**Technical Novelty And Significance:** 1
**Empirical Novelty And Significance:** 1
**Recommendation:** 1

**Clarity, Quality, Novelty And Reproducibility:**

Paper is written in a very convoluted way and it is very hard to follow what exactly is being proposed new in the model in the paper that didn't already exist in the literature.

Experiments section gives not enough details about the runs to really understand what exactly was done for fitting. Also, no detail about where the cases data for UAE is taken is given in the paper.

**Strength And Weaknesses:**

Strengths: -
 * Idea of having compartments for pre-symptomatic and symptomatic populations is necessary if you want to model SARS-CoV-2
 * dataset used is a real-world data of the epidemic
 * Model shows th established properties like having R0 < 1 for controlling the epidemic

Weakness: -
* The paper is not self-contained it is very cumbersome to actually get the whole idea behind the proposed model
* Experiments section is written in a way that it is not obvious to know how experiments are conducted and how can you really get that model is well calibrated. For example, it is hard to know what Figures 4-7 are actually showing and how is that any validation of the model.
* the claim of using pre-symptomatic and symptomatic as two new compartments as novel is incorrect. Almost all established models already have these. For example see [1].
* Paper has wrongfully claimed that they are first ones with stochastic model for SARS-CoV-2. Again, many of the models cited in paper are stochastic and the model in [1] is much more powerful than proposed by authors.
* Fitting to only 3 months of data that too only on cases is not that much useful

[1] Sonabend, R., Whittles, L. K., Imai, N., Perez-Guzman, P. N., Knock, E. S., Rawson, T., ...Cori, A. (2021). Non-pharmaceutical interventions, vaccination, and the SARS-CoV-2 delta variant in England: a mathematical modelling study. Lancet, 398(10313), 1825–1835. doi: 10.1016/S0140-6736(21)02276-5

**Summary Of The Paper:**

Authors present an extension to an already existing model for modelling dynamics of SARS-CoV-2 spread by accounting for pre-symptomatic and symptomatic populations. They present a real-world case study example to support their proposed model.

**Summary Of The Review:**

Please look above.

---

### Official Review · Reviewer_Xoxa · 2022-11-01

**Confidence:** 3
**Correctness:** 2
**Technical Novelty And Significance:** 1
**Empirical Novelty And Significance:** 2
**Recommendation:** 3

**Clarity, Quality, Novelty And Reproducibility:**

Novelty

The novelty of the proposed approach seems rather limited. In particular, while previous approaches had not considered stochasticity in modeling transitions, the population partitions were already characterized. Further, standard techniques are used to derive the update equations and prove the desirable qualities of the model. While the derivations appear quite technical to a non-expert such as myself, they appear to largely be based on standard methods.

Quality

In terms of technical quality, I did not verify the derivations in detail, but they appear correct.

I did not find the experimental results to be convincing. As mentioned above, they do not make clear to what extent the method and results generalize. First, the evaluation is limited to a relatively short period of time at a single location. Thus, it is not clear if the model would generalize to other locations which may have, for example, different immigration patterns or different masking policies.

Second, the predictive ability of the model is not assessed. So we do not know to what extent the model would work within the same location at different points in time, such as in winter, when the behavior patterns of individuals may change.

Third, the proposed approach misses the beginning of the spike in the time and location on which it was tested. Presumably, this would be the most important trend to pick up since it would suggest some change in policy would be needed.

Clarity

The writing has a number of small typos. While I do not believe they affect understanding of the paper, they do become distracting. The paper would benefit from another round of editing.

The references are not consistently formatted.

As a small clarification for the beginning, MERS is the result of a merbecovirus, not a sarbecovirus; they are all betacoronaviruses, though.

The notation in the paper is quite heavy, but the authors do a reasonable job of explaining the intuition behind the parapers. I also appreciate that they found space for a notation table.

Reproducibility

As far as I could tell, no code or synthetic datasets were provided. Nevertheless, I believe similar results could be obtained based on the equations given in the paper.


**Strength And Weaknesses:**

The main strength of the paper lies in adding stochasticity to transitions among the population partitions in an existing model.

Another strength of the paper is the description and intuition given for the various parameters introduced in the equations throughout the text.

As described in more detail below, though, both the novelty and technical quality of the work is limited. The technical quality would be significantly improved by demonstrating the generalizability of  the model using a data-driven approach. Such an evaluation may also help improve the novelty, in the sense that it could help show the benefit of the stochasticity.

**Summary Of The Paper:**

In this work, the authors propose a differential equation approach to model the temporal dynamics of SARS-Cov-2 in a single location. The model incorporates stochasticity in modeling the transitions among five partitions of the population. They also use standard analytic techniques to show that the proposed model is well-posed. Finally, a limited set of experiments suggests the model can model some of the dynamics involved in one small dataset.

**Summary Of The Review:**

Overall, I believe the limited methodological contributions and lack of experimental evidence of generalizability will limit the impact of this work. Also, considering its strong emphasis on epidemiological modeling, it may interest a broader audience at a venue catered to that topic.

---

### Decision · Program_Chairs · 2023-01-20

**Decision:**

Reject

**Justification For Why Not Higher Score:**

The reviewers were uniform in their judgment.

**Justification For Why Not Lower Score:**

N/A

**Metareview: Summary, Strengths And Weaknesses:**

The paper proposes an SDE model for SARS-CoV-2 transmission with pre-symptomatic and symptomatic compartments. While the model proposed is reasonable, there is significant room for the paper to be improved. The model needs to be situated in the relevant literature, and the claimed novelty put into perspective given the vast array of compartmental models that have already been proposed. The reviewers also noted the lack of content relevant to a machine learning audience, so I would recommend seeking a different venue in the future.